# The pattern of orthopedic fractures and visceral injury in road traffic crash victims, Addis Ababa, Ethiopia

**Zuriyash Mengistu**[1]*, **Ahmed Ali**[2], **Teferi Abegaz**[2]

**1** School of Nursing and Midwifery, College of Health Sciences, Addis Ababa University, Addis Ababa, Ethiopia, **2** School of Public Health, College of Health Sciences, Addis Ababa University, Addis Ababa, Ethiopia

* zuriyashaau@yahoo.com

**Data Availability Statement:** All relevant data are within the manuscript and its Supporting Information files.

**Funding:** Addis Ababa University provide financial support for this study. The funders had no role in

## Abstract

### Background

Road Traffic crash injury is one of the main public health problems resulting in premature death and disability particularly in low-income countries. However, there is limited evidence on the crash fractures in Ethiopia.

### Objective

The study was conducted to assess the magnitude of road traffic crash fractures and visceral injuries.

### Methods

A hospital-based cross-sectional study was conducted on 420 fracture patients. Participants were randomly selected from Addis Ababa City hospitals. The study was carried out between November 2019 and February 2020. Data were collected using a questionnaire and record of medical findings. Multilevel logistic regression analysis was carried out. Ethical clearance was obtained from the Addis Ababa University, College of Health Sciences Institutional Review Board. Confidentiality of participants' information was maintained.

### Results

The study found out that the majority 265 (63. 1%) of fracture cases were younger in the age group of 18 to 34 years. Males were more affected—311(74.0%). The mortality rate was 59 (14.1%), of those 50(85.0%) participants were males. The major road traffic victims were pedestrians—220(52.4%), mainly affected by simple fracture type -105(53.3%) and compound fracture type—92(46. 7%). Drivers mainly suffered from compound fracture type -23 (59.0%). One hundred eighty-two (43.3%) of fracture patients had a visceral injury. Homeless persons who sit or sleep on the roadside had a higher risk of thoracic visceral injury compared to traveler pedestrians (AOR = 4.600(95%CI: 1.215–17.417)); P = 0.025.

study design, data collection and analysis, decision to publish, or preparation of the manuscript.

**Competing interests:** no competing interest exist.

**Abbreviations:** IRB, Institutional Review Board; RTI, Road Traffic Injury; WHO, World Health Organization; USA, United State of America.

## Conclusion

Visceral injury, simple and compound fractures were the common orthopedic injury types reported among crash victims. Males, pedestrians, and young age groups were largely affected by orthopedic fracture cases. Homeless persons who sited or slept on the roadside were significant factors for visceral injury. Therefore, preventing a harmful crash and growing fracture care should be considered to reduce the burden of crash fracture.

## Introduction

Road Traffic crash has been a public health problem leading to fatal and serious injuries [1]. According to the World Health Organization [WHO] 2018 Report, there were 1.35 million annual road crash deaths with an additional 20 to 50 million serious injuries in the world [2]. Since the numbers of road deaths are the tip of the iceberg, the report could not show serious injuries and their consequences. Likewise, there has been no reduction in the number of road traffic deaths and serious injuries in low-income countries since 2013 [2]. Accordingly, it is necessary to study the magnitude of orthopedic fracture patterns [3]. A fracture can happen when the applied physical force on the bone is stronger than the bone strength. Studies showed that road traffic crash is the commonest cause of orthopedic fractures [4, 5]. The study revieled that crash injury can be characterized interms of ordinal feature of injury severity [6].

Males are almost three times more affected by orthopedic fractures compared to females [7]. Different studies found out frequently more affected by road traffic crashes than females [8–14]. The younger age groups are most commonly affected in orthopedic crash fractures [1, 10, 11, 15–17]. The area of fracture also varies among travelers. Unbelted drivers and passengers are largely affected by head and face injuries [18]. Studies stated that people with low socioeconomic status had been exposed to lower-limb fracture or a spine/trunk fracture [8, 19]. Younger age groups are more vulnerable due to their involvement in productive activities that require them to move fast enough from one area to another and due to their risky behaviors [5, 16, 17]. The majority of crash victims are pedestrians who are mostly affected by extremity fracture because of direct impact without any protection [7].

The most common bone fractures by body region are fractures of the lower limbs and visceral injury of pelvic, head, intra-abdominal and thoracic regions [19]. The isolated most common bone fractures occur on the femur [1] and the tibia/fibula [20, 21]. Fracture patterns are frequently associated with other injuries especially head injuries [8]. Another study revealed that shoulder and wrist dislocations are the most common dislocations [11]. In Addis Ababa, at least one crash death is recorded daily and 28,361 crashes caused 4,433 human injuries in 2019 [22].

In Ethiopia, studies about orthopedic fracture patterns and visceral injuries due to road traffic crashes are limited. Therefore, the purpose of this study was to assess the pattern of orthopedic fractures and associated visceral injuries due to crashes.

## Materials and methods

### Study area and period

This study was conducted in Addis Ababa City Administration which is the Capital City of Ethiopia. Addis Ababa City is the seat of the African Union (AU) and the United Nations Economic Commission for Africa (UNECA). Addis Ababa was founded by Emperor Minilik II

and Empress Taitu in 1887. The history of the City 's road development also started at the inception of the City. Emperor Minilik II is the first king to import two cars to Addis Ababa and to introduce car technology in the City for the first time in 1907 [23]. In 2019, Ethiopia had 935,888 vehicles of which 60% were in Addis Ababa [22].

This study was conducted between November 2019 and February 2020 among randomly selected tertiary level hospitals in Addis Ababa City. Those hospitals were Tikur Anbessa Specialized Hospital, AaBET Hospital, and Minilik II Hospital.

## Study design

We conducted a cross sectional study in the three randomly selected tertiary hospitals where crash victims had arrived at emergency departments of the respective hospitals.

## Population

Victims of road traffic crash injury who diagnosed as orthopedic fracture based on radiologic findings by examining medical doctors constituted the source population. The study included every individual orthopedic patient with age of greater or equal to 18 years old, having musculoskeletal road traffic crash fracture that require tertiary level hospitals care.

Orthopedic fractured patients who died during pre-hospital time, and who were referred or transferred to other hospitals were excluded.

## Sample size determination

Sample size was calculated using a single population proportion formula ($n = Z^2\alpha–½ p(1 –p) / d^2$). The following assumptions were considered for sample size determination; (i) proportion of most frequently affected rib fracture-associated injury on chest (31.0%%) [9]; (ii) 95% confidence level of discrepancy rate of 5%; (iii) a design effect of 1.5. Accordingly, a total of 493 study subjects were calculated for this study.

## Sampling technique and procedure

All tertiary care hospitals that provide trauma service in Addis Ababa were identified and listed. The three Hospitals (Minilic II Hospital, Tikur Anbessa Hospital, and AaBET Hospital) that provide service to orthopedic victims were randomly selected. The study subjects were recruited based on eligibility criteria to collect demographic and crash-related primary data and fracture-related secondary data from the selected hospitals' emergency departments.

We included all eligible orthopedic fracture patients in the random selection sampling procedure.

## Data collection instrument and procedure

Data collection tools included a structured questionnaire to be responded to by victims of orthopedic fracture. Besides, patient records data that had radiographic orthopedic fracture results and orthopedic fracture medical diagnosis were utilized. The data for this study were obtained from selected injured patients after fracture diagnosis and who received emergency care. Data were collected by assigned emergency medical doctors and emergency graduate nurses at emergency departments.

The training was given for two days for data collectors and supervisors about the objectives of the study and how to collect data.

The contents of the questionnaire were extracted from the relevant literature review and pretested. The injury severity score criteria: mild injury and above were established.

Correlation analysis was done to eliminate correlated explanatory variable such as passenger and vehicle collision with a roadside object, the significant factor passenger retained in the injury severity study.

The Questionnaire was translated from English to Amharic and again back to English to ensure consistency. The pre-test of the data collection tool was done in Zewditu Hospital. The supervisors and investigator facilitated the data collection process. The principal investigator worked with data collectors to assure the trustworthiness of the data and to minimize inter-observer bias. Every day, the collected data were checked for completeness for missing value and consistency using the check-list. Double data entry was done for coded data using STATA 14 to ensure accuracy.

## Data processing and analysis

Data were cleaned for outliers. The validated data were prepared for analysis. In this study, descriptive analysis; frequency, range, and median were computed. Multilevel logistic regression was employed to differentiate individual and group level risk factors for visceral injury. The assumption of multilevel analysis is the variation in the dependent variable at one level explained as a function of variables defined at various levels. Thus, multilevel analysis allows researchers to deal with the micro-level of individuals and the macro-level of groups simultaneously.

**Estimation.** We use Restricted Maximum Likelihood (REML) which is the preferred and typically the default method, because of variance unbiased with small samples. Estimate assumes a normal distribution of all random effects. All parameters estimated (fixed effects, random group effects, variances of the random effects, and residual variance) were simultaneously estimated using iterative methods.

**Parameter testing.** For the test of variance–Wald test, and test of fixed effects- t-tests were used.

Model fit—the fit of the model is given by the model's "deviance?" called, "– 2LogLikelihood "in SPSS. The deviance of different nested models can be compared using a $X^2$ test [24].

The following steps are used for statistical analysis.

Level 1 Model (individual level)

$$Yij = \beta0 + \beta1 * (\text{Sex}) + \ldots\ldots\ldots + 4 * (\text{simple fracture}) + \varepsilon ij \qquad (2)$$

Level 2 Model (cluster level)

$$\beta0 = \gamma00 + \gamma01 * (\text{Pedestrians}) + \ldots\ldots + \gamma04 * (\text{homeless persons}) + Uoj \qquad (3)$$

To estimate injury severity of traffic crash injury,

The study used OP(Ordered Probit) to model injury severity, since it enable for identifying statistically significant relationships between explanatory variables and dependent variable. It also requires smaller samples when compared to unordered response models, besides it discerns unequal differences between ordinal classes in the dependent [25, 26].

The Ordered probit model is used for caracterizind the orderd nature of injury sveity. The model is built around the notion of a latent underlying injury risk propensity occurring from vehicle crash that determines the observed ordinal injury severity level. Hence, this study follows the path of Kockelman and Kweon [25, 26].The following specification was used here:

$$Zi = Xi\beta + \varepsilon_i \qquad \text{Eq(1)}$$

where Zi denotes the latent injury risk propensity for crash victim i, β is the vector of parameters to be estimated, Xi is the vector of observed non-random explanatory variables measuring the attributes of crash victim i, and εi is the random error term following standard normal distribution.

Accordingly, the mean and the variance of εi are normalized to zero and one, respectively. Since the dependent variable, Zi, is unobserved, standard regression techniques cannot be applied to compute Eq (1). Therefore, one can reasonably assume that a high risk of injury, denoted by Zi is related to a high level of observed injury, denoted by Yi [25–27]. This relationship can be translated as follows

$$Yi = \begin{cases} 1, & \text{if } Zi \leq \mu1 \\ k, & \text{if } \mu k - 1 < Zi \leq \mu k \\ K, & \text{if } Zi > \mu k - 1 \end{cases} \quad \text{Eq(2)}$$

Where $\mathbf{\mu} = \{\mathbf{\mu}1, \ldots, \mathbf{\mu}_k, \ldots, \mathbf{\mu}k\text{ -}1\}$ are the threshold values for all injury severity levels that define Yi corresponding to integer ordering, and K is the highest ordered injury severity level. In turn, the probability that accident victim i faces an injury severity level k is equal to the probability that the latent injury risk propensity, Zi, assumes a value between two fixed thresholds. In other words, given the value of Xi, the probability that the injury severity faced by accident victim i belongs to each injury severity level is:

$$\begin{cases} P(y = 1) = \Phi(-\beta X_i) \\ P(y = k) = \Phi(\mu_{k-1} - \beta X_i) - \Phi(\mu_{k-2} - \beta X_i) \\ P(y = K) = 1 - \Phi(\mu_{K-1} - \beta X_i) \end{cases} \quad \text{Eq(3)}$$

where ɸ is the cumulative normal distribution function. To estimate Eq (3), The formula written as = >

$$P(y = k) = \Phi(\mu_k - \beta X_i) - \Phi(\mu_{k+1} - \beta X_i) \quad \text{Eq(4)}$$

where μk and μk +1 denote the lower and upper thresholds for the injury severity level k, respectively.

These marginal effects provide the direction of the probability for each level as follows:

$$P(y = k)/\partial X = \lfloor \Phi(\mu_k - \beta X_i) - \Phi(\mu_{K-1} - \beta X_i) \rfloor \beta \quad \text{Eq(5)}$$

All statistical analysis was done using STATA and Statistical Package for Social Science (SPSS).

## Ethical consideration

Ethical clearance was obtained from the Addis Ababa University, College of Health Science Institutional Review Board (IRB) with Protocol number: 036/17/SPH, before conducting the study. Permission to undertake the study was obtained from officials in the selected hospital setting. Since all patients cannot read and write crash victims and/or surrogate offered oral informed consent for free study participation according to IRB approved consent procedure. The verbal cnsent was documented in patient's medical record. Complying with a patient's request, the collected data would be removed from the database. Confidentiality of information about orthopedic fracture patients was maintained.

## Result

### Socio-demographic characteristics of orthopedic fractured patients

A total of 420 road traffic crash victims with orthopedic fractures consented and agreed to participate in this study with an 86% response rate. The majority of victims were males 311

(74.0%), giving a male to female ratio of 3:1 (Table 1). The age range was from 18 to 85 years old (median 29 years). The majority of respondents, 265(63.1%,) were in the young age group (18 to 34 years). Regarding the marital status of patients, almost similar proportions of married 213 (50.7%) and unmarried 207 (49.3%) cases were noted. The educational levels of

**Table 1. Demographic and crash-related characteristics of traffic crash victims, Ethiopia, 2020.**

| Variable | Frequency | Percentage (%) |
|---|---|---|
| Sex: | | |
| Male | 311 | 74.4 |
| Female | 109 | 25.6 |
| Age groups | | |
| 18–24 | 138 | 32.8 |
| 25–34 | 127 | 30.2 |
| 35–44 | 75 | 17.9 |
| 45–54 | 39 | 9.3 |
| >54 | 41 | 9.8 |
| Marital status: | | |
| Married | 213 | 50.7 |
| Unmarried | 207 | 49.3 |
| Religion: | | |
| Orthodox | 303 | 72.2 |
| Muslim | 58 | 13.8 |
| Protestant | 56 | 13.3 |
| Catholic | 3 | 0.7 |
| Education: | | |
| Illiterate | 57 | 13.6 |
| Primary school | 103 | 24.5 |
| Secondary school | 178 | 42.4 |
| Higher education | 82 | 19.5 |
| Road participation | | |
| Pedestrian | 197 | 46.9 |
| Driver | 39 | 9.3 |
| Passenger | 162 | 38.6 |
| Homeless person (sitting /sleeping on road side) | 22 | 5.2 |
| Seat belt use | | |
| Yes | 14 | 35.0 |
| No | 26 | 65.0 |
| Mortality | | |
| Alive | 361 | 85.9 |
| Death | 59 | 14.1 |
| Mechanism | | |
| ** Pedestrian direct hit by a vehicle | 212 | 50.5 |
| Rollover | 62 | 14.8 |
| Collusion | 111 | 26.4 |
| Strike with roadside object | 29 | 6.9 |
| Rejecting | 6 | 1.4 |

* Homeless pedestrians (sitting /sleeping roadside)

** Traveler pedestrian strike by vehicles (due to over speed, brake failure, drinking)

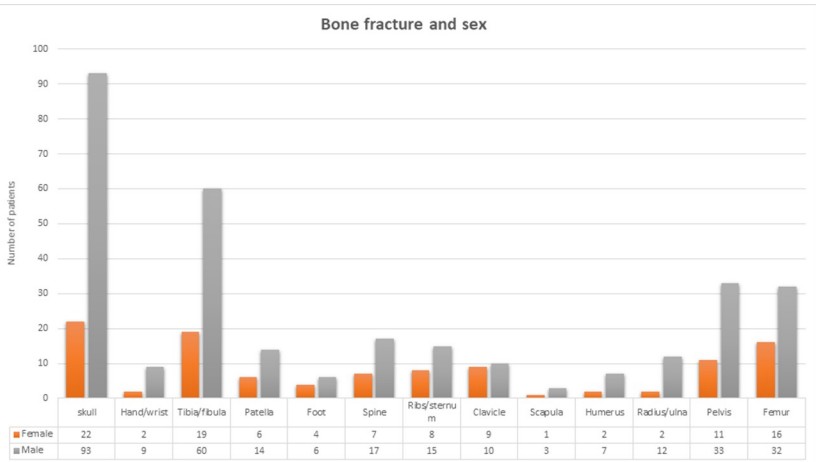

**Fig 1. Sex category and isolated fracture sustained among road crash victims, Ethiopia, 2020.**

participants were dominantly high school complete, 178(42.4%). Nearly half of orthopedic fracture cases were pedestrians 197(46.9%) followed by car occupants 162(38.6%) and drivers 39(9.3%). All seat belt user patients were among drivers 14(54.4%).

Out of total fracture cases accounting for 59(14.1%), victims died, of those 50(85.0%) were males. Regarding road user fatal cases, pedestrians accounting for nearly two-thirds, 37(62.7%) died. Considering the mechanism of the crash, crash fractures were mainly resulting from vehicle strike with the pedestrian, 212(50.5%) cases followed by vehicle–vehicle collision crash 111(26.4%) cases.

## Bone fracture type concerning road user characteristics

Based on the results shown in Fig 1 males had a higher prevalence of all categories of fracture than females. The leading fractured bones in males and females were the skull accounting for 93 (29.9%) in males and 22 (20.2%) in females, followed by the tibia/fibula 60(19.3%) in males and 19 (17.4%) in females. In males, the pelvic bone fracture and femurs bone fracture occurred almost in equal proportions 33 (7.9%) and 32 (7.6%) respectively. However, in females, the skull and the tibia/fibula bone fractures occurred in almost equal proportions 22 (5.2%) and 19(5,0%) respectively.

Regarding the type of fracture, more than half of pedestrian travelers, 105(53.3%) had a simple fracture and the rest 92(46.7%) had a compound fracture (Table 2). Likewise, almost

**Table 2. Fracture type and the number of fractures among traffic crash victims, Ethiopia, 2020.**

| Variables | Pedestrian | Driver | Passenger | Others * |
|---|---|---|---|---|
| | Frequency (%) | Frequency (%) | Frequency (%) | Frequency (%) |
| Type of Fracture | | | | |
| Simple fracture | 105(53.3) | 16(41.0) | 87(53.7) | 15(68.2) |
| Compound fracture | 92(46.7) | 23(59.0) | 75(46.3) | 7(31.8) |
| Number of fracture: | | | | |
| Single bone fracture | 85(43.2) | 9(23.0) | 89(55.0) | 14(63.6) |
| Two bone fracture | 53(26.9) | 18(46.2) | 36(22.2) | 3(13.6) |
| Multiple bone fracture | 59(29.9) | 12(30.8) | 37(22.8) | 5(22.7) |

Others *(Homeless person who sleep or sit on the roadside)

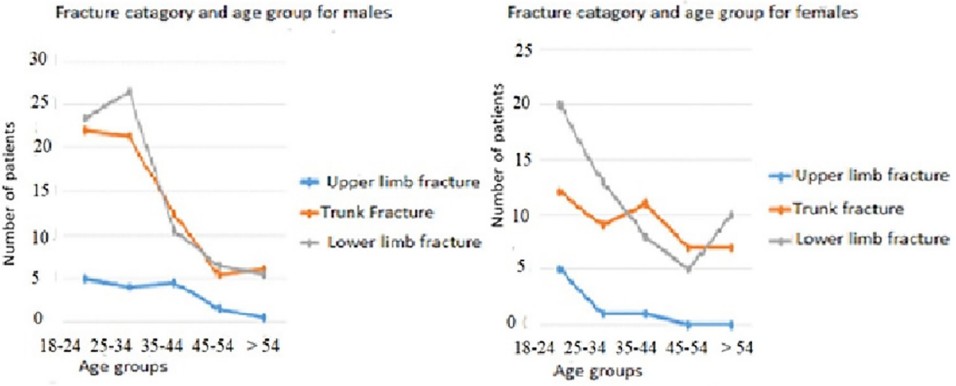

**Fig 2. Fracture category and age groups in males and females among road crash victims, Ethiopia, 2020.**

the same proportion of passengers, 87(53.7%) and 75(46.3%) experienced a simple fracture and compound fracture respectively. On the other hand, drivers had largely compound bone fractures in 23(59.0%) cases. In our study, passengers mainly had a single fracture, 89(55.0%) followed by multiple fractures 37(22.8%). The fracture patterns for drivers were double fracture in 18(46.2%) cases, followed by multiple fractures in 12(30. 8%) cases. Homeless pedestrians who sit or sleep on the roadside had simple 15(68.2%) and single 14(63.6%) fractures.

Fig 2 depicts the distributions of three orthopedic fracture patterns (upper limb, trunk, and lower limb). The fracture pattern of lower limb and trunk among males had two 'peaks', through the age of 18–24 and 25–34 years of age, and progressively lower through 35 to 44 years.

In females, the fracture pattern of lower limb and trunk showed two 'peaks', in the age group of 18–34 years and on those ages greater than 54 years, and was lower through ages of 35 to 45 years, and those demonstrated virtually U-shape pattern. Females also had upper limb fracture 'peak' through the age of 18 to 24 years and decreased fracture frequency in those aged greater than 45 years.

## Magnitude and site of dislocation

This study found out that 29 (6.9%) of bone fractured patients had associated dislocation. Among dislocated orthopedic cases, the shoulder joint was the leading dislocated joint found in 9 (31.0%) cases, followed by hip joint dislocation found in 8 (28.6%) cases. The elbow, wrist/finger, and ankle joint dislocation were affected in equal proportion, of two(6.9%)in each case (Table 3).

## Visceral injury

Our study revealed that nearly half of bone fractured patients had visceral injury in 182(43.3%) cases (Table 3). The commonest visceral injury was head injury 104 (24.8%), followed by pelvic and genitourinary injury 38(9.0%). The smaller number of bone fractured patients had abdominal visceral injuries 8(1.9%).

## Associated factors for visceral injury

A multilevel analysis was used to evaluate the individual effects (i.e. male sex, age 18–29 years, compound type of fracture, and the group effect (roadside sleeping or sitting type of homeless

**Table 3. Joint dislocation and visceral injury of road traffic crash victims, Ethiopia, 2020.**

| Joint dislocation | Frequency | Percentage (%) |
|---|---|---|
| Without dislocation | 391 | 93.1 |
| Jaw | 0 | 0 |
| Shoulder | 9 | 2.1 |
| Elbow | 2 | 0.5 |
| Wrist/finger | 2 | 0.5 |
| Hip | 8 | 1.9 |
| Knee | 6 | 1.4 |
| Ankle | 2 | 0.5 |
| Total | 420 | 100 |
| Associated Visceral Injury | | |
| Without visceral injury | 238 | 56.7 |
| Head injury | 104 | 24.8 |
| Thoracic injury | 32 | 7.6 |
| Abdominal injury | 8 | 1.9 |
| Pelvic and genitourinary injury | 38 | 9.0 |

road user) on the visceral injury of orthopedic fracture cases compared to no visceral injury fracture cases (Table 4).

The study revealed that being female sex orthopedic patient has a significantly lower risk of visceral injury of the head that decreased by 51%t compared to the male patient (AOR = 0.492 (95%CI: 0,275- .880); $P < 0.017$).

Among educational level factors, being highly educated victims had a highly significant risk factor in the visceral injury of the head (AOR = 3.846(95%CI:1.580–9.363); P = 0.003).

Compared with the age stratum of 18–29 years, increased ages (age greater than 54 years) had significantly reduced the risk of visceral injury of the head by 67% (AOR = 0.327 (95% CI:0.113–0.949); P<0.04). Among the type of fracture factor compound fracture was noted to be a significant risk factor in the visceral injury of the abdomen- pelvic (AOR = 1.952(95% CI:1.016–3.748); P = 0.045). A road user group factor clustered to homeless orthopedic patients who sit or sleep on the roadside had a significantly higher risk of thoracic visceral injury compared to traveler pedestrian groups (AOR = 4.600(95%CI: 1.215–17.417)); P = 0.025.

## Road traffic crash injury severity estimates using an ordered probit model

**Dataset characterization.** Data related to on vehicle and out vehicle crashes have been sorted out to develop the model. The observed injury severity sustained by orthopedic crash victims is distributed as follows: slightly injured (19.1%); seriously injured (68.7%); and fatal injury (12. 2%). Accordingly, the dependent variable considered in this study is the level of injury severity sustained by orthopedic crash victims which divided into three categorical levels: slightly injured (y = 1); seriously injured y (y = 2); and fatal injury (y = 3). The explanatory variables (independent variables) comprises the variable related to crash victim and collusion type. Table 5 presents the definition of each explanatory variable together with its mean (M) and standard deviation (SD) values. All of these variables, with the exception of the variable age of the orthopedic crash victims are binary with means between 0 and 1. The variable age (continuous variable) has been scaled (dividing by 100) to have mean with the same scale as those of the binary, since Ordered Probit (OP) models may not converge if the variables have not similar scales (25–27).

**Table 4. Multilevel logistic regression of factors for visceral injury of crash victims, Ethiopia, 2020.**

| Independent factors | Type of visceral injury | | | | | |
| --- | --- | --- | --- | --- | --- | --- |
| | Head injury | | Thoracic injury | | Abdomen- pelvic injury | |
| | P-value | AOR (95%CI) | P-value | AOR (95%CI) | P-value | AOR(95%CI) |
| Sex* | | | | | | |
| Male | | 1.0 | | 1.0 | | 1.0 |
| Female | 0.017 | 0.492(0,275- .880) | 0.105 | 0.437(0.161–1.189) | 0.708 | 1.138(0.578–2.242) |
| Age* | | | | | | |
| 18–24 | | 1.0 | | 1.0 | | 1.0 |
| 25–34 | 0.696 | 0.866(0.420–1.784) | 0.589 | 0.808(0.371–1.757) | 0.220 | 1.611(0.751–3.457) |
| 35–44 | 0.645 | 0.820(0.353–1.907) | 0.761 | 0.870(0.352–2.147) | 0.522 | 1.337(0.549–3.260) |
| 45–54 | 0.445 | 1.517(0.520–4.428) | 0.287 | 1.860(0.593–5.835) | 0.470 | 1.528(0.482–4.838) |
| >54 | 0.040 | 0.327(0.113–0.949) | 0.901 | 0.933(0.313–2.779) | 0.901 | 0.933(0.313–2.779) |
| Education* | | | | | | |
| Illiterate | | 1.0 | | 1.0 | | 1.0 |
| Primary school | 0.196 | 1.772(0.743–4.225) | 0.810 | 0.860(0.252–2.936) | 0.547 | 1.382(0.481–3.968) |
| Secondary school | 0.3240 | 1.505(0.667–3.397) | 0.641 | 0.767(0.251–2.343) | 0.527 | 0.716(0.253–2.024) |
| Higher education | 0.003 | 3.846(1.580–9.363) | 0.361 | 1.769(0.519–6.031) | 0.061 | 2.785(0.953–8.135) |
| Type of fracture* | | | | | | |
| Simple fracture | | 1.0 | | 1.0 | | 1.0 |
| Compound fracture | 0.565 | 0.873(0.548–1.390) | 0.340 | 0.690(0.322–1.481) | 0.045 | 1.952(1.016–3.748) |
| Road user group **: | | | | | | |
| Pedestrian | | 1.0 | | 1.0 | | 1.0 |
| Driver | 0.895 | 1.054(0.480–2.317) | 0.643 | 1.380(0.352–5.403) | 0.114 | 0.192(0.025–1.493) |
| Passenger | 0.740 | 1.089(0.657–1.804) | 0.120 | 1.960(0.838–4.584) | 0.824 | 0.926(0.468–1.832) |
| Others*** | 0.753 | 1.198(0.388–3.703) | 0.025 | 4.600(1.215–17.417) | 0.304 | 1.917(0.553–6.649) |

*Individual-level.

**Cluster level.

***Others (Homeless person who sit or sleep on the roadside)

This study finds out the results of variables estimation and the marginal effects model of crash victims. The coefficient of each variable represents how it affects the severity of the injury. The positive sign of a coefficient means that the corresponding variable tends to

**Table 5. Description of explanatory variables among orthopedic crash victims, Ethiopia,2020.**

| Explanatory variables | Description | Mean | SD |
| --- | --- | --- | --- |
| Sex | 1 if male, 0 if female | 0.741 | 0.438 |
| Age | Continuous variable | 0.329 | 0.143 |
| Road participation: | | | |
| Out vehicle road participant | 1 if pedestrians 0 if not pedestrians | 0.524 | 0.500 |
| On vehicle road participant | 1 if passenger 0 if not passenger | 0.479 | 0.500 |
| Mechanism of crash | | | |
| Direct hitting | 1 if direct hit 0 if not direct hit | 0.505 | 0.501 |
| Rolling | 1 if rolling 0 if not rolling | 0.162 | 0.369 |
| vehicle to vehicle collusion | 1 if vehicle to vehicle 0 if not vehicle to vehicle | 0.264 | 0.442 |
| striking with road side object | 1 if striking with road side object 0 if not striking with road side object | 0.069 | 0.254 |

**Table 6. Traffic crash injury severity model estimates of ordered probit among road traffic crash victims, Ethiopia, 2020.**

| Variable | Estimation Parameter | P-value | Marginal effects | | |
|---|---|---|---|---|---|
| | | | Slight injury | Serious injury | fatal injury |
| Sex: | | | | | |
| Male | 0.27335 | 0.02 | -0.76316 | 0.02735 | 0.04896 |
| Age | 0.72952 | 0.05 | - 0.19299 | 0.05209 | 0.14091 |
| Road participation: | | | | | |
| Out vehicle road participant | - 0.45963 | 0.71 | 0.12073 | - 0.03095 | - 0.08977 |
| On vehicle road participant | - 0.43252 | 0.72 | 0.11476 | - 0.03142 | - 0.08334 |
| Collision type: | | | | | |
| Vehicle direct hit the victims | 0.59280 | 0..04 | -0.15547 | 0.03933 | 0.116143 |
| Rollover | 0.06665 | 0.78 | - 0.01727 | 0.00409 | 0.01321 |
| Vehicle to Vehicle | 0.30494 | 0.20 | - 0.07553 | 0.01155 | 0.06397 |

increase the severity level compared to the default variable. In contrast, coefficients that have negative signs tend to decrease the severity level of injury as the corresponding variable increases.

The marginal effects indicate the independent effects of explanatory variables on the changes in the probability of having a certain level of injury.

From the 16 independent variables observed, sex and collusion type are variables influencing the crash victims' fatality. Crash victims have a higher possibility of dying in accidents if the sex is men, and collusion type occurred between vehicle to vehicle(Table 6).

The coefficient value shows that if the standard deviation of men's sex changes by 1 level, the crash victim's severity of dying will increase as much as 27.0% when other variables remain the same. If the standard deviation of vehicle-to-vehicle collision type changes by 1 level, the severity of crash victims of dying will increase as much as 31.0% when the other variables remain the same.

## Discussion

The study found out that the younger age groups (18–34 years) were the main victims of road traffic crash fracture. This finding is in agreement with studies conducted in India (15–30 years], and Saudi Arabia (21-30-years) [1], where the majority of road traffic victims were young adults. The possible explanation could be that younger groups are energetic and fast in manipulating vehicles and also take risks. The orthopedic fracture was common among males (74.0%). This finding is in agreement with studies conducted in Iran (77%)], German (76%) [9], and Uganda (71%)]. This could be due to more involvement of males in out-of-door activities, greater exposure of traffic, violation speed limit, driving under alcohol intoxication, impatience, and risk behavior than females.

Half of the fractured pedestrians (50.5%) experienced crashes resulting from pedestrians hit by vehicles. The study participants witnessed that the hit by vehicle occurred due to vehicle over speed, brake failure, and drink deriving. Thus, it is important to monitor the drivers' alcohol level and driving efficiency along with vehicle safety compliance. Males died six-fold compared to females. It was higher compared to the study conducted in Taiwan nearly three-fold in that study]. This might be due to the limited involvement of females in-vehicle driving and limited use of road mobility by females in Ethiopia. Out of total mortality (14.1%), the mortality of vehicle occupants and vulnerable road users were 4.1% and 8.8% respectively. The mortality rate was higher than a prospective study conducted in the United Arab Emirates where mortality had been 2% among vehicle occupants and 4% among vulnerable road users [15].

These differences could be due to differences in the study design and the infrastructure and driver awareness. In this study, fourteen drivers (35.0%) used a seat belt. Our result was in contrast to the study that showed 94% of drivers had fastened their seat belts]. The possible explanation might be due to the loose enforcement of seat belt use in our setting.

As per our study, the leading isolated bones fractured that occurred in males (29.9%) and females (20.2%) were the skull. This finding is in contrast with other findings, where humerus, radius/Ulna, Tibia, and Tibia/fibula orthopedic fracture injuries were the common isolated fractures in Saud Arabia], India], Cameron], and USA [21] respectively. The possible suggestion might be due to pedestrian crash exposure without protection in our study. In this study, the least commonly fractured bone in both males and females was the scapula. This is in agreement with the study conducted in Saudi Arabia].

In particular, pedestrian road users had simple (53.3%) and compound (46.7%) fractures. These findings are consistent with the study conducted in Uganda]. The possible explanations for the large numbers of pedestrian fractures might be mainly due to mixing of pedestrians walking and vehicles running on the same road, lack of awareness of drivers and pedestrians, and lack of compliance with highway regulations. The occurrence of simple and compound fractures in pedestrians was almost similar. This could be due to pedestrians experiencing crashes from a direct hit by vehicles. Thus, depending on the intensity of a direct hit collision, the pedestrian could experience a simple or compound fracture. Drivers mainly sustained compound bone fractures (59.0%), with injury patterns involving two bone fractures (46.2%). The possible explanation could be that the majority of drivers (52.5%) had vehicle-vehicle collusion followed by rollover (32.5%). Further, large proportions of drivers (65.0%) were unbelted that result in less protection from forced collusion.

During early age, males and females showed 'peaks' in lower limb and trunk fractures that progressively declined throughout middle age. As age advances, fractures remain lower in males but fracture peak returned in females. This may suggest that at an early age males and females participated in the road to travel to school and to the workplace that increases traffic exposure. But in advanced age the decrease in bone density might be higher in females than males, leading to fracture vulnerability. This study depicted that the shoulder joint was the most commonly dislocated bone joint whereas the least commonly affected joint was the Ankle joint. This finding is in agreement with studies conducted in India].

This study found out that nearly half of orthopedic patients (43.3%) had visceral injuries. The most common visceral injury was brain injury (24.8%). Several factors including the individual effect (i.e. male sex, age 18–29 years, compound type of fracture) and the group effect (homeless roadside slept type of road-users) were associated with the visceral injury. Specifically, our study revealed that being higher education level victims had a highly significant risk for visceral injury of the head. This might be due to among road participants higher educated victims were mainly on vehicle traveler (53.7%) who confronted a large proportion of viscera injury of the head (56.7%) from the total of all type of visceral injury. The possible assumption could be on vehicle travelers practice lower use of seat belts and a reasonably functional vehicle equipped with a protective device. The study revealed that being female orthopedic patients had a significantly lowered risk of visceral injury of the head by 51%t compared to male patients (AOR = 0.492(95%CI: 0,275- .880); $P < 0.017$). This could be due to lower risky behavior of females, not incurring serious law-breaking faults and a limited number of female travelers. The study revealed that both men and women were able to understand and detect risk, but only women showed concern and high attention about the risk of driving [13]. Males have frequently shown adventurous and aggressive traits and their tendency to drive in risky behaviors including trespassing speed limits, reckless overtaking, non-use of seatbelts, and driving under the influence of alcohol and impatience [13, 14].

Increased ages of greater than 54 years had reduced the risk of visceral injury of the head by 67% (AOR = 0.327 (95%CI:0.113–0.949); P<0.04). This suggests that the decreased bone density in advanced age may reduce the bone force exerted on the brain. The collusion intensity might be decreased due to advanced age's calmness to control the vehicle speed, traffic negotiation, risky behavior, and impatience. Studies indicate that road traffic crash occurs ultimately due to high traffic volume and infringement of safety violations unless driver minimize the hazard with impotence and negotiation [16] but, young drivers are more likely to underestimate the risk of being involved in a crash and to overestimate their abilities as drivers. Young males drivers are more prone to accept speeding, traffic violations, and drugs and alcohol use by the driver [13]. Another study revealed the most common risky behaviors like driving on the sidewalk, answering a phone call, and driving in fatigue [17].

Compound fractures were noted to be a significant risk factor in the visceral injury of the abdomen- pelvic (AOR = 1.952(95%CI: 1.016–3.748); P = 0.045). This may be because the excessive force is exerted on the bone. The higher educational level had been a significantly higher risk for visceral injury of the head. However, there were a small number of higher educated participants 28(6.7%) to give a meaningful interpretation for that finding. Homeless orthopedic patients who sleep or sit on the roadside had a higher risk for thoracic visceral injury compared to traveler pedestrian groups (AOR = 4.600(95%CI: 1.215–17.417)); P = 0. 025. This may be due to homeless victims in roadside sitting and sleeping positions may decrease their visibility by the vehicle driver.

The study shows factors such as sex, and vehicle-to-person collision type was found to be significantly associated with injury severity.

The study had the inevitable limitation. First, the patients who die during transportation, transfer, and referred may lower the orthopedic fracture burden. Second, collecting data at the time of hospital arrival may not be inclusive to collect the overall situation of scene time and transport time characteristics of crash injured victims. Third, this study did not cover the long-term disability of orthopedic fractured patients.

Finally, this study did not consider medical and surgical treatment outcomes of orthopedic fractured cases.

## Conclusion and recommendation

In conclusion, orthopedic fracture, dislocation, and visceral injury were the commonest road traffic injuries posing considerable mortality and morbidity in our study. Head injury and lower limb fractures were more common. We also observed that males, the homeless pedestrian who sit or sleep on the roadside, and those aged greater than 54 years were the most affected orthopedic patients.

It could be concluded that men's sex and vehicle to vehicle collision type can increase the fatality of road traffic crash victims.

Therefore; road traffic crash as a public health problem deserves considerable attention to mitigate the looming burden.

Government agencies should enforce regulations like proper seatbelt use, put road signs to keep travelers and drivers attentive on-road environment.

It is also important to focus on the younger age group to reduce risky behavior to minimize crash fractures.

Males need to be considered for crash fracture reduction by using protecting device for crash exposure and by participating females in driving activity to large extent.

The community should have mobilized for awareness at the maximum level by targeting the high-risk group including homeless victims who sit or sleep on the roadside, younger age groups, and males to be prevented and protected from crash fracture.

Health managers should consider the site of fractures like extremity and associated visceral injury such as skull and abdomen which occur in common to overcome further impairment.

Furthermore, we hope that for researchers, this study on the prevalence of orthopedic fractures, joint and visceral injuries by body site will provide a baseline for further studies, by targeting the high-risk group such as younger age groups, males, the homeless person who sit or sleep on roadside and traveler pedestrians, and long term orthopedic crash fracture disabilities.

## Supporting information

**S1 Dataset.**
(DOCX)

## Acknowledgments

We would like to thank the late Prof. Fikre Enquselassie for his contribution during the development of the proposal.

Our appreciation goes to the Tikur Anbessa, Minilic II, and ABaT hospitals' management committees, staff, and data collectors for their support during the data collection process.

## Author Contributions

**Conceptualization:** Zuriyash Mengistu.

**Data curation:** Zuriyash Mengistu.

**Formal analysis:** Zuriyash Mengistu.

**Investigation:** Zuriyash Mengistu.

**Methodology:** Zuriyash Mengistu.

**Supervision:** Ahmed Ali, Teferi Abegaz.

**Writing – original draft:** Zuriyash Mengistu.

**Writing – review & editing:** Ahmed Ali, Teferi Abegaz.

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
