## [Decision Letter · Decision Letter 0]

16 Sep 2020

PONE-D-20-25631

Pattern of orthopedic fractures and visceral injury among road traffic crash victims, Addis Ababa , Ethiopia.

PLOS ONE

Dear Dr. assen,

Thank you for submitting your manuscript to PLOS ONE. After careful consideration, we feel that it has merit but does not fully meet PLOS ONE’s publication criteria as it currently stands. Therefore, we invite you to submit a revised version of the manuscript that addresses the points raised during the review process.

Please address all the reviewers' comments and improve the manuscript.

We look forward to receiving your revised manuscript.

Kind regards,

Quan Yuan, Ph.D.

Academic Editor

PLOS ONE

Journal Requirements:

2. Please include additional information regarding the survey or questionnaire used in the study and ensure that you have provided sufficient details that others could replicate the analyses. For instance, if you developed a questionnaire as part of this study and it is not under a copyright more restrictive than CC-BY, please include a copy, in both the original language and English, as Supporting Information.  If the original language is written in non-Latin characters, for example Amharic, Chinese, or Korean, please use a file format that ensures these characters are visible.

3. Please state whether you validated the questionnaire prior to testing on study participants. Please provide details regarding the validation group within the methods section.

4. Please amend your current ethics statement to address the following concerns:  

a) Did participants provide their written or verbal informed consent to participate in this study?

"No"

6. Please amend either the title on the online submission form (via Edit Submission) or the title in the manuscript so that they are identical.

Reviewers' comments:

Reviewer's Responses to Questions

**Comments to the Author**

1. Is the manuscript technically sound, and do the data support the conclusions?

Reviewer #1: Yes

Reviewer #2: Yes

2. Has the statistical analysis been performed appropriately and rigorously? 

Reviewer #1: Yes

Reviewer #2: Yes

3. Have the authors made all data underlying the findings in their manuscript fully available?

Reviewer #1: Yes

Reviewer #2: Yes

4. Is the manuscript presented in an intelligible fashion and written in standard English?

Reviewer #1: No

Reviewer #2: Yes

5. Review Comments to the Author

Reviewer #1: This paper investigates the orthopedic fractures and visceral injury among road traffic crash victims in Addis Ababa, Ethiopia using data collected from questionnaire and medical records. A multilevel logistic model is used to identify their influencing factors. The research topic is interesting and worth of investigation. The paper is generally well organized. However, in the paper, more works on analysis of traffic injury severity based logit/probit models should be acknowledged, such as:

Investigating the impacts of real-time weather conditions on freeway crash severity: A Bayesian spatial analysis. International Journal of Environmental Research and Public Health, 2020, 17(8), 2768.

Analyzing freeway crash severity using a Bayesian spatial generalized ordered logit model with conditional autoregressive priors. Accident Analysis and Prevention, 2019, 127, 87-95.

The interactive effect on injury severity of driver-vehicle units in two-vehicle crashes. Journal of Safety Research, 2016, 59: 105-111.

Besides, the English writing should be improved, as there are some grammar mistakes in the manuscript.

Reviewer #2: The manuscript shows an interesting and useful topic from the aspect of traffic safety. Hospital based cross sectional multilevel logistic regression analysis was carried out on 420 fracture patients. Finally, the patterns of orthopedic fractures and related visceral injuries caused by traffic accidents were evaluated. However, there are several issues and details need to be addressed. The detailed comments are as follows:

1. The figures and tables in the manuscript are not standardized and unified, and the list of references should be consistent with the requirements of the journal.

2. The authors did not pay attention to the grammatical mistakes and conciseness of the paper, there are many sentences that are too long and wordy. It is suggested that the author should read it carefully and repair it.

3. This paper obtains the data through the survey questionnaire, but lacks the reliability and validity test of the questionnaire quality. The rationality of questionnaire design will directly affect the analysis value of survey data and the understanding of orthopedic fractures and related visceral injuries caused by traffic accidents patterns by traffic departments, and then affect road construction, management of drivers and formulation of policies and regulations. In order to ensure the accuracy and reliability of the final evaluation results. Please add two indicators of reliability and validity to test the design quality of the questionnaire.

4. Page 26, “Several factors were significant in explaining which factors are more associated with visceral injury using multilevel analysis”. What are the specific factors? In the discussion section, It is suggested that the author should add list and explain which factors are related to visceral injury instead of a brief statement.

5. Also on page 26, “The collusion intensity might be decreased due to advanced age’s calmness to control the vehicle speed, traffic negotiation, risky behavior and impatience”. Please add more reasonable explanation or evidence from previous studies to confirm this result.

6. PLOS authors have the option to publish the peer review history of their article (what does this mean?). If published, this will include your full peer review and any attached files.

Reviewer #1: No

Reviewer #2: No

---

## [Decision Letter · Decision Letter 1]

26 Jan 2021

PONE-D-20-25631R1

Pattern of orthopedic fractures and visceral injury in road traffic crash victims, Addis Ababa, Ethiopia.

PLOS ONE

Dear Dr. assen,

Thank you for submitting your manuscript to PLOS ONE. After careful consideration, we feel that it has merit but does not fully meet PLOS ONE’s publication criteria as it currently stands. Therefore, we invite you to submit a revised version of the manuscript that addresses the points raised during the review process.

Please address all the reviewer's concern and revise the manuscript again.

We look forward to receiving your revised manuscript.

Kind regards,

Quan Yuan, Ph.D.

Academic Editor

PLOS ONE

Reviewers' comments:

Reviewer's Responses to Questions

**Comments to the Author**

1. If the authors have adequately addressed your comments raised in a previous round of review and you feel that this manuscript is now acceptable for publication, you may indicate that here to bypass the “Comments to the Author” section, enter your conflict of interest statement in the “Confidential to Editor” section, and submit your "Accept" recommendation.

Reviewer #1: (No Response)

Reviewer #2: All comments have been addressed

2. Is the manuscript technically sound, and do the data support the conclusions?

Reviewer #1: (No Response)

Reviewer #2: Yes

3. Has the statistical analysis been performed appropriately and rigorously? 

Reviewer #1: (No Response)

Reviewer #2: Yes

4. Have the authors made all data underlying the findings in their manuscript fully available?

Reviewer #1: (No Response)

Reviewer #2: Yes

5. Is the manuscript presented in an intelligible fashion and written in standard English?

Reviewer #1: (No Response)

Reviewer #2: Yes

6. Review Comments to the Author

Reviewer #1: The specification of the ordered Logit model is vague. There are some errors in the formulations. The authors are suggested to revise this part in accordance with the previous works.

Reviewer #2: The manuscript shows an interesting and useful topic from the aspect of traffic safety. Hospital based cross sectional multilevel logistic regression analysis was carried out on 420 fracture patients. Finally, the patterns of orthopedic fractures and related visceral injuries caused by traffic accidents were evaluated. The paper has carried on the concrete revision and the reply according to the comments.

7. PLOS authors have the option to publish the peer review history of their article (what does this mean?). If published, this will include your full peer review and any attached files.

Reviewer #1: No

Reviewer #2: No

---

## [Decision Letter · Decision Letter 2]

6 Apr 2021

PONE-D-20-25631R2

The Pattern of orthopedic fractures and visceral injury in road traffic crash victims, Addis Ababa, Ethiopia.

PLOS ONE

Dear Dr. assen,

Thank you for submitting your manuscript to PLOS ONE. After careful consideration, we feel that it has merit but does not fully meet PLOS ONE’s publication criteria as it currently stands. Therefore, we invite you to submit a revised version of the manuscript that addresses the points raised during the review process.

Please address all the reviewer's concern and revise the manuscript again.

We look forward to receiving your revised manuscript.

Kind regards,

Quan Yuan, Ph.D.

Academic Editor

PLOS ONE

Journal Requirements:

Reviewers' comments:

Reviewer's Responses to Questions

**Comments to the Author**

1. If the authors have adequately addressed your comments raised in a previous round of review and you feel that this manuscript is now acceptable for publication, you may indicate that here to bypass the “Comments to the Author” section, enter your conflict of interest statement in the “Confidential to Editor” section, and submit your "Accept" recommendation.

Reviewer #1: (No Response)

Reviewer #2: All comments have been addressed

2. Is the manuscript technically sound, and do the data support the conclusions?

Reviewer #1: (No Response)

Reviewer #2: Yes

3. Has the statistical analysis been performed appropriately and rigorously? 

Reviewer #1: (No Response)

Reviewer #2: Yes

4. Have the authors made all data underlying the findings in their manuscript fully available?

Reviewer #1: (No Response)

Reviewer #2: Yes

5. Is the manuscript presented in an intelligible fashion and written in standard English?

Reviewer #1: (No Response)

Reviewer #2: Yes

6. Review Comments to the Author

Reviewer #1: There are still some confusions on the specification of the ordered logit model. How is the latent variable Y_*_i linked the observed crash injury severity? The authors are suggested to refer to more works on this model.

Reviewer #2: The manuscript shows an interesting and useful topic from the aspect of traffic safety. Hospital based cross sectional multilevel logistic regression analysis was carried out on 420 fracture patients. The paper has carried on the concrete revision and the reply according to the comments.

7. PLOS authors have the option to publish the peer review history of their article (what does this mean?). If published, this will include your full peer review and any attached files.

Reviewer #1: No

Reviewer #2: No

---

## [Decision Letter · Decision Letter 3]

11 Jun 2021

The Pattern of orthopedic fractures and visceral injury in road traffic crash victims, Addis Ababa, Ethiopia.

PONE-D-20-25631R3

Dear Dr. assen,

We’re pleased to inform you that your manuscript has been judged scientifically suitable for publication and will be formally accepted for publication once it meets all outstanding technical requirements.

Kind regards,

Quan Yuan, Ph.D.

Academic Editor

PLOS ONE

Additional Editor Comments (optional):

Reviewers' comments:

Reviewer's Responses to Questions

**Comments to the Author**

1. If the authors have adequately addressed your comments raised in a previous round of review and you feel that this manuscript is now acceptable for publication, you may indicate that here to bypass the “Comments to the Author” section, enter your conflict of interest statement in the “Confidential to Editor” section, and submit your "Accept" recommendation.

Reviewer #1: All comments have been addressed

Reviewer #2: All comments have been addressed

2. Is the manuscript technically sound, and do the data support the conclusions?

Reviewer #1: (No Response)

Reviewer #2: Yes

3. Has the statistical analysis been performed appropriately and rigorously? 

Reviewer #1: (No Response)

Reviewer #2: Yes

4. Have the authors made all data underlying the findings in their manuscript fully available?

Reviewer #1: (No Response)

Reviewer #2: Yes

5. Is the manuscript presented in an intelligible fashion and written in standard English?

Reviewer #1: (No Response)

Reviewer #2: Yes

6. Review Comments to the Author

Reviewer #1: (No Response)

Reviewer #2: The manuscript shows an interesting and useful topic from the aspect of traffic safety. Hospital based cross sectional multilevel logistic regression analysis was carried out on 420 fracture patients. Finally, the patterns of orthopedic fractures and related visceral injuries caused by traffic accidents were evaluated. The paper has carried on the concrete revision and the reply according to the comments.

7. PLOS authors have the option to publish the peer review history of their article (what does this mean?). If published, this will include your full peer review and any attached files.

Reviewer #1: No

Reviewer #2: No

---

## [Editor Report · Acceptance letter]

26 Aug 2021

PONE-D-20-25631R3 

The pattern of orthopedic fractures and visceral injury in road traffic crash victims, Addis Ababa, Ethiopia.  

Dear Dr. Mengistu:

I'm pleased to inform you that your manuscript has been deemed suitable for publication in PLOS ONE. Congratulations! Your manuscript is now with our production department. 

Kind regards, 

on behalf of

Dr. Quan Yuan 

Academic Editor

PLOS ONE